# Practical behavioural solutions to COVID-19: Changing the role of behavioural science in crises

**Charlotte C. Tanis**[1]\*, **Floor H. Nauta**[1], **Meier J. Boersma**[2], **Maya V. Van der Steenhoven**[2], **Denny Borsboom**[1], **Tessa F. Blanken**[1]\*

**1** Department of Psychological Methods, University of Amsterdam, Amsterdam, The Netherlands, **2** Smart Distance Lab, Leiderdorp, The Netherlands

\* c.c.tanis@uva.nl (CCT); t.f.blanken@uva.nl (TFB)

**Data Availability Statement:** All contact data is publicly available on figshare: Tanis, C.C.; Blanken, Tessa (2022): Contact data supermarket.

## Abstract

For a very long time in the COVID-19 crisis, behavioural change leading to physical distancing behaviour was the only tool at our disposal to mitigate virus spread. In this large-scale naturalistic experimental study we show how we can use behavioural science to find ways to promote the desired physical distancing behaviour. During seven days in a supermarket we implemented different behavioural interventions: (i) rewarding customers for keeping distance; (i) providing signage to guide customers; and (iii) altering shopping cart regulations. We asked customers to wear a tag that measured distances to other tags using ultra-wide band at 1Hz. In total $N = 4,232$ customers participated in the study. We compared the number of contacts (< 1.5 m, corresponding to Dutch regulations) between customers using state-of-the-art contact network analyses. We found that rewarding customers and providing signage increased physical distancing, whereas shopping cart regulations did not impact physical distancing. Rewarding customers moreover reduced the duration of remaining contacts between customers. These results demonstrate the feasibility to conduct large-scale behavioural experiments that can provide guidelines for policy. While the COVID-19 crisis unequivocally demonstrates the importance of behaviour and behavioural change, behaviour is integral to many crises, like the trading of mortgages in the financial crisis or the consuming of goods in the climate crisis. We argue that by acknowledging the role of behaviour in crises, and redefining this role in terms of the desired behaviour and necessary behavioural change, behavioural science can open up new solutions to crises and inform policy. We believe that we should start taking advantage of these opportunities.

## Introduction

Behaviour is often mentioned in relation to crises: the *trading* of mortgages that resulted in the financial crisis in 2008, the *shaking* of hands in the latest COVID-19 crisis, or the *consuming* of goods in the climate crisis. In most of these cases, behaviour is primarily considered as a factor that causes or sustains a crisis, but when it comes to solving a crisis, behaviour is less likely to be considered. For solutions to a crisis, we often turn to experts from the respective discipline

University of Amsterdam / Amsterdam University of Applied Sciences. Dataset. https://doi.org/10.21942/uva.20052083.v1.

**Funding:** The research project was supported by the Ministry of Economic Affairs and Climate Policy. CT and TB were supported by an Innovation Exchange Amsterdam UvA Proof of Concept Fund. The funders had no role in study design, data collection and analysis, decision to publish, or preparation of the manuscript.

**Competing interests:** The authors have declared that no competing interests exist.

—in the financial crisis we turn to economists [1], in the COVID-19 crisis we turn to epidemiologists [2], and in the climate crisis we turn to climatologists [3]. We seem to overlook the power of behavioural change, and the field of behavioural science, in our battle against these crises [4]. We should consider people's behaviour not only as causes and sustaining factors, but also turn to the solutions that behavioural change can provide.

The most recent COVID-19 crisis clearly demonstrates the role of behaviour and behavioural change [5, 6]. The virus transmitted through behavioural contacts and it was pivotal that we found ways to alter people's behaviour and promote physical distancing and hygiene to mitigate virus spread [7]. This need for behavioural change resulted in numerous studies into factors that determine behaviour [8–10], and worldwide regulations such as lockdowns, school closures, and travel restrictions, all directed to reduce the number of behavioural contacts. Interestingly, while these studies and regulations focused on behavioural change, their effectiveness was assessed in terms of psychological constructs such as intentions and motivations and epidemiological parameters like the reproduction number, number of cases and deaths [11]. Behavioural criteria to express the effectiveness of these regulations in directly observed behaviour (e.g., to what extent did people keep their distance) were largely unavailable [12, 13] and there was little information on whether these interventions did also successfully accomplish the desired behavioural changes. The lack of direct assessments is especially problematic given the discrepancy between people's intentions and motivations and their actual behaviour, also known as the intention-behaviour gap [14, 15].

We have seen similar patterns in other crises. Take for example the global financial crisis of 2008, where behaviour clearly played a role, e.g., through the creation of hedge funds and taking excessive risks [16]. In finding a solution, however, governments were inclined to turn to economical solutions and e.g., lowered the interest rates [17]. At the same time, there are important questions that need to be asked, such as why did people engage in this excessive risk taking behaviour, and how could other behaviour be stimulated? [18] Such questions are crucial to understand and prevent other financial crises, and are intrinsically of behavioural nature. In the climate crisis too, behaviour is considered as cause while predominantly technological solutions are being proposed. However, it has been increasingly vocalized that the only way to combat climate change is through behavioural change [19, 20]. These examples underscore the importance of behaviour and behavioural change, not only as causing and sustaining factors of crises, but precisely also as solutions to crises.

Behavioural science offers many models into the determinants of behaviour, and thereby offers leveraging points on how to instantiate behavioural change. The widely used Capability, Opportunity, Motivation, Behaviour (COM-B) model, for example, posits that a particular behaviour occurs when someone has the *capability* (i.e., psychological and physical), *opportunity* (i.e., contextual factors that facilitate the behaviour), and *motivation* (i.e., processes that energize and direct behaviour) to enact the behaviour [21]. By outlining the factors that influence particular behaviours to occur, the model also provides the opportunity to identify different points of engagement to bring about the desired behavioural change.

We advocate that behavioural science plays a prominent role in finding solutions to crises, *together* with scientists from other fields [22]. In the context of the COVID-19 crisis, we need scientists from many different disciplines [23]: virologists and micro-biologists to understand how the virus works [24], epidemiologists on how the virus spreads [25], and medical scientists on how to treat the virus [26]. But we *also* need behavioural scientists to understand how we can successfully change our behaviour and combat the virus spread. Behavioural science provides a way to link science and society, with behavioural change running like a thread out of crises.

In the current paper we demonstrate how we can put this idea into practice, and use behavioural science to provide concrete answers on how to promote the desired behaviour of physical distancing during the COVID-19 crisis, similar to our previous work in an art fair [27]. In this research we focused on physical distancing behaviour in public spaces, specifically in a supermarket. We identified different ways to stimulate physical distancing by considering psychological processes, crowd management, and practical solutions (see Methods for details): (i) rewarding customers; (ii) providing signage to guide customers; and (iii) altering shopping cart regulations. Importantly, we directly measured the desired behaviour using wearable sensors that recorded the distance between customers. We subsequently systematically evaluated the effectiveness of these interventions on physical distancing in an experimental design where we varied the proposed interventions across seven days. The goal of this research was to use behavioural science to help find concrete solutions to the COVID-19 crisis by clearly defining the desired behaviour (physical distancing), implementing behavioural interventions to stimulate this behaviour, and using a direct and objective measurement of this behaviour to assess its effectiveness.

## Materials and methods

### Participants and design

Participants in our naturalistic study were customers of the supermarket PLUS André and Joyce van Reijen in Veldhoven, the Netherlands. Veldhoven is a town of approximately 45,000 inhabitants in the southern Netherlands, located in the Metropoolregio Eindhoven. All customers older than 16 years could participate in the study, and there were no other in- or exclusion criteria.

The experiment took place during seven days in a supermarket. We varied three interventions: reward [28], signage [29], and adjusting the shopping cart regulations. These interventions were chosen for varying reasons. First, rewarding people for displaying the desired behaviour is well-established to be effective in promoting that desired behaviour [30], and also advised during the COVID-19 pandemic in particular [31]. In our study, participants received the reward upon handing in their tag (see Procedure) and consisted of cookies on Saturday March 24[th] and chocolate on Friday March 26[th]. Second, signage is commonly used to change behaviour [29] and often used in traffic to, for example, avoid collisions [32]. We aimed to investigate whether clear signage would facilitate pedestrian flows and thereby physical distancing. We included arrows signaling unidirectional walking directions in part of the supermarket and footprints in the queue for the register (see Materials below). Third, we changed the shopping cart regulations from mandatory (which was standard in the Netherlands at that time) to optional. The mandatory shopping carts were implemented in the Netherlands in March 2020 [33] to (1) keep track of the number of participants in the supermarket as national regulations allowed a maximum of 1 customer per 10 m$^2$, and (2) as the shopping carts were thought to facilitate physical distancing. At the same time it could be argued that the mandatory shopping carts take up a lot of space within the supermarket and, as such, hinder the opportunity for physical distancing.

We varied the interventions across days, resulting in a unique set of interventions for each day, see Table 1. Note that there is some redundancy in the experimental conditions we implemented: for example, the effect of reward could be assessed by comparing day four to day three or two, but also by comparing day 7 to day 6. We did so to minimise the influence of factors such as day and time of day. Both the day and time of day are likely to affect the type of customers and crowdedness in the supermarket. Customers during workdays may differ from customers during the weekend, and customers during the morning may differ from customers

**Table 1. Experimental design.**

| Day | | Intervention | | | | Comparison |
|---|---|---|---|---|---|---|
| | | Reward | Signage | | Shopping cart | |
| | | | Arrows | Footprints | | |
| 1 | 17 March 2021 | | | | | shopping cart |
| 2 | 18 March 2021 | | | | ✓ | |
| 3 | 19 March 2021 | | | | ✓ | shopping cart, signage |
| 4 | 20 March 2021 | ✓ | | | ✓ | |
| 5 | 24 March 2021 | | | ✓ | | |
| 6 | 25 March 2021 | | ✓ | ✓ | ✓ | signage, reward |
| 7 | 26 March 2021 | ✓ | ✓ | ✓ | ✓ | reward |

after work-hours. Similarly, the crowdedness in the supermarket is likely to affect the number of contacts made. Since these factors are difficult to control in a naturalistic study, we implemented some redundancy in the design so that we could select time intervals post-hoc that would minimise differences in these factors for an optimal comparison across conditions.

## Materials

**Physical distance.** Participants wore a SafeTag developed by KINEXON (https://kinexon.com/technology/safetag/). SafeTags are wearable tags that measure the distance to other Safe-Tags at a frequency of one Hz using ultra-wideband (UWB) technology with an accuracy up to 10 cm. A tag automatically turns on and starts measuring when taken out of its charging unit, and turns back off when placed back into the unit. The SafeTags measure the distance to all other tags that are active, and it is technically not possible to link multiple SafeTags when they belong to members of the same group, so to exclude contacts made between group members. Each tag has a unique tag id and locally stores measured distances until the data is read out on a laptop running the management software. The tags were solely used to measure physical distance, and did not provide any form of feedback.

**Shopping experience.** We asked participants to rate their shopping experience on an iPad. Participants were asked to rate three questions on a five-point scale from satisfied to dissatisfied (indicated by five emoticons): (1) How did you experience the corona regulations in the supermarket? ('regulations'); (2) How pleasant was it to do groceries like this? ('pleasantness'); (3) Did you feel you were helped to keep a distance? ('help').

**Camera and traffic light.** National regulations specified a maximum number of customers inside the supermarket (i.e., one customer per 10 $m^2$). The mandatory shopping cart allowed to keep track of the number of participants inside. In order to be able to relax the mandatory shopping cart but still adhere to national regulations, we installed a camera at the entrance of the supermarket that counted all incoming customers. The camera was linked to a traffic light that indicated to incoming customers how crowded the supermarket was (green: few people inside; orange: quite busy, be aware of your distance; red: full capacity reached, wait until someone exits). The camera registered all incoming and outgoing customers which was saved to a database and could be retrieved per hour.

**Signage.** We had two types of signage in the supermarket: waiting signage in the queue for the register and arrows depicting unidirectional walking directions in part of the supermarket where the aisles were too narrow to keep 1.5 m distance. The waiting signage consisted of round stickers with footprints on them and were placed 1.5 m from one another. The arrows

were also placed 1.5 m from one another, to both signal the unidirectional walking directions as well as the appropriate physical distance.

## Procedure

In the week prior to our experiment (week 10, 2021), flyers were passively distributed alongside the ad brochure in the supermarket, to inform customers of the upcoming experiment on the effectiveness of physical distancing regulations. The flyer informed participants when the study would take place, and stressed its voluntary character. See S1 Appendix for the flyer.

The experiment itself took place in week 11 from Wednesday March 17th until Saturday March 20th and week 12 from Wednesday March 24th until Friday March 26th 2021. Each day we handed out sensors between 12:00 and 17:00. The supermarket communicated via posters that every day between 14:00 and 15:00 was intended for older and vulnerable people to do their groceries, but this was not actively enforced.

Outside of the supermarket, a team member informed customers that a study would take place inside the supermarket, for which they could participate on a voluntary basis. On days that shopping carts were not mandatory, posters were present that informed customers that they could enter without taking a shopping cart. Inside, customers were asked to participate in the experiment. In case customers had additional questions about the study, a team member took the customer aside to avoid queuing and explained the study in more detail and provided the original information flyer. If customers agreed to participate, we handed them a SafeTag to wear on a lanyard around their neck, and registered their implicit informed consent. If customers of the same household participated, they each received a SafeTag. According to regulations, household members did not need to keep a distance, but we were unable to register their group membership upon handing out the tag to avoid congestion. We did exclude contacts between group members when processing the data and only kept contacts with individuals outside of the group (see Pre-processing).

In conditions in which a reward was handed out, we also informed participants that they would receive a reward for their effort to keep their distance upon handing in their SafeTag. Participants could then proceed to do their groceries like they would otherwise do. Thus, as soon as the participants entered the supermarket they did not encounter any study personnel. After paying at the register, participants could rate their satisfaction with the supermarket visit. A desk to hand back the tags was located at the exit of the supermarket. Both the tags and lanyards were thoroughly cleaned before they were handed out again to other customers.

The ethics review board of the University of Amsterdam approved the study and implicit informed consent because of the voluntary and anonymous nature of the study (2021-PML-13247).

## Analysis

**Pre-processing.** The raw data from the SafeTags contains for each assessment (frequency of 1 Hz) a time stamp, the tag id of the reporting and opposing tag, and the distance between them in centimetres. Since tags were handed out multiple times during one day, we first determined the start and end time of each participant wearing the tag to construct unique participant id's. We then checked if participants entered the supermarket as a household by investigating three criteria: at least 10 contacts within 80 cm, being within 1.5 m of each other for at least 25% of their visit duration, and exiting the supermarket at most 60 seconds after each other. We assigned participants as belonging to the same group if they met at least two of these criteria and removed all contacts between the respective group members. Note that any contacts to other participants present in the supermarket were retained. We considered two

participants to be in contact with one another if (a) they did not belong to the same group and (b) were within 1.5 m from one another, in accordance with the physical distance regulations at the time in the Netherlands. To compute the contact duration between two customers, we summed the duration of all registered contacts between two participants.

**Descriptive analyses.** We first performed a simple linear regression to assess the effect of crowdedness on the median number of unique contacts per hour. As was found in other studies [34] it is conceivable that the busier it is, the more difficult it becomes to keep a physical distance. If this is the case, then we should control for this effect by comparing conditions at times that were similar in crowdedness.

To evaluate the effectiveness of each intervention in isolation, we select two conditions that differ only in regard to whether the behavioural intervention of interest is implemented and that are similar in crowdedness and time of day. This way we can isolate the effect of the intervention while keeping other factors constant. For each intervention, we tested the difference in the number of unique contacts between participants with a Bayesian logistic regression model (see Contact networks), and in contact duration with a Mann-Whitney U test. In addition, we tested differences in ratings of shopping experience (i.e., regulations, pleasantness, and help) with two sample t-tests using all available data from that day.

**Contact networks.** To compute differences in number of contacts between participants, we analyzed the contact networks with a Bayesian logistic regression model, developed for our previous study, called the b2 model [27]. The b2 model is a reduced version of the multilevel p2 model [35], for undirected (i.e., a contact is always shared between two people) and unweighted (i.e., a contact is binary and duration is not taken into account) networks. The model omits reciprocity parameters, dyadic predictors, and random effects at the network level, and contains identical random sender and receiver effects. We modeled differences in the number of unique contacts between two contact networks with actor-level dummy variables. Estimating the b2 model was done in a similar manner as the j2 model [36, 37], using Markov Chain Monte Carlo simulation and similar prior distributions. For all comparisons, we report the posterior means as point estimates for the odds ratios accompanied by the corresponding 95% credible interval. The credible interval describes the range where the true odds ratio lies with 95% certainty.

All analyses were performed in R (version: 4.1.1) and the `dyads` (1.1.4) package [38] was used to compare the contact networks.

## Results

### Visitors

A total of $N = 4,232$ customers participated in our study. The number of customers inside the supermarket varied over time, and we first investigated whether there was an effect between crowdedness and the number of contacts per hour. Fig 1 shows that there exists a relation between the number of customers inside as registered by the camera at the entrance, and the median number of contacts per hour ($F(1, 28) = 10.89$, $p = .003$, $R^2 = .28$). To minimize the influence of sample size and customer type on the effects of interventions, we selected comparable hours in terms of the time of day, number of customers inside the supermarket, and number of customers participating in our study (compliance), see Table 2. This resulted in a selection of six time slots distributed over four days.

### Contact network

To evaluate the effectiveness of behavioural interventions on physical distancing we followed the experimental framework proposed by Blanken et al. (2021) [27] in which participants and

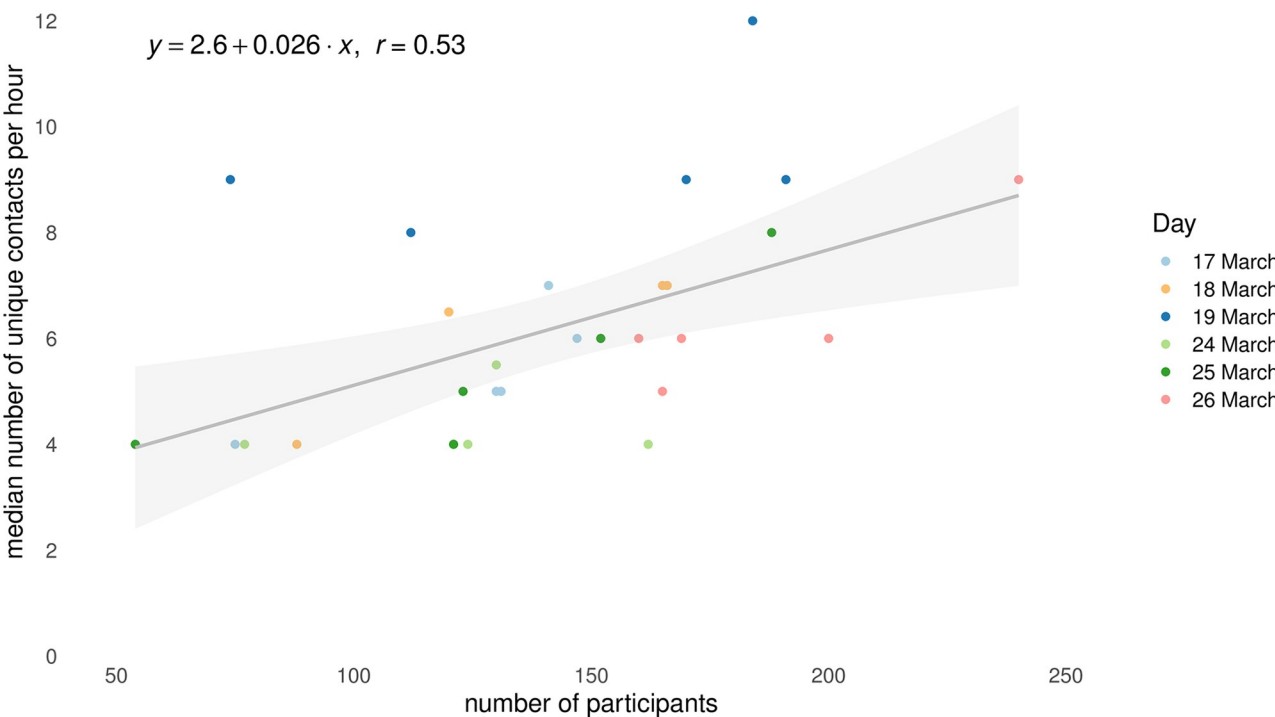

$y = 2.6 + 0.026 \cdot x, \ r = 0.53$

Day
- 17 March
- 18 March
- 19 March
- 24 March
- 25 March
- 26 March

**Fig 1. Crowdedness and contacts.** Relationship between the number of participants and the median number of unique contacts in one-hour time windows across six days.

their contacts are represented in a *contact network*. This representation allows to take the network structure into account when comparing two conditions [12].

Fig 2 shows the contact network of the $n = 624$ participants included on the first day of our study. Each participant is represented as a node, and whenever two participants came within 1.5 m of each other, they are connected by a link. The highlighted nodes indicate $n = 147$ participants present between 15:00 and 16:00 that were included in the analysis.

**Table 2. Descriptives.**

| Condition | | Day | | $n_{reg}$ | $n_{part}$(%) | Number of contacts | | | | Contact duration | | | | Experience | | | |
|---|---|---|---|---|---|---|---|---|---|---|---|---|---|---|---|---|---|
| | | | | | | Range | $M \pm SD$ | Median | IQR | Range | $M \pm SD$ | Median | IQR | $n$ | Regulation | Pleasantness | Help |
| reward | no | 6 | 16–17h | 275 | 188 (68%) | 0–24 | 8.4 ± 5.3 | 8 | 4–12 | 2–22 | 7.0 ± 4.0 | 5.9 | 4.0–8.5 | 240 | 4.3 ± 0.8 | 4.0 ± 0.9 | 4.1 ± 0.9 |
| | yes | 7 | 15–16h | 316 | 200 (63%) | 0–26 | 6.7 ± 5.1 | 6 | 3–9 | 2–28 | 5.6 ± 4.1 | 4.2 | 3.2–6.6 | 238 | 4.2 ± 0.8 | 4.0 ± 0.9 | 4.1 ± 0.8 |
| signage | no | 3 | 15–16h | 237 | 170 (72%) | 0–33 | 9.5 ± 6.3 | 9 | 4–13 | 2–44 | 6.8 ± 4.7 | 5.6 | 4.3–8.0 | 194 | 4.1 ± 0.9 | 3.7 ± 1.0 | 3.9 ± 0.9 |
| | yes | 6 | 15–16h | 222 | 152 (68%) | 0–27 | 6.1 ± 4.5 | 6 | 3–8 | 2–36 | 7.0 ± 5.9 | 4.8 | 3.6–8.0 | 240 | 4.3 ± 0.8 | 4.0 ± 0.9 | 4.1 ± 0.9 |
| shopping carts | mandatory | 1 | 15–16h | 204 | 147 (72%) | 0–27 | 7.7 ± 5.7 | 6 | 4–10 | 2–48 | 7.4 ± 5.9 | 6.0 | 4.0–8.7 | 324 | 4.2 ± 0.8 | 3.8 ± 1.0 | 3.9 ± 1.0 |
| | optional | 3 | 15–16h | 237 | 170 (72%) | 0–33 | 9.5 ± 6.3 | 9 | 4–13 | 2–44 | 6.8 ± 4.7 | 5.6 | 4.3–8.0 | 194 | 4.1 ± 0.9 | 3.7 ± 1.0 | 3.9 ± 0.9 |

Table notes $n_{reg}$ indicates the number of incoming customers, as registered by the camera at the entrance, and $n_{part}$ indicates the number of customers who agreed to wear a tag and participate in the study.

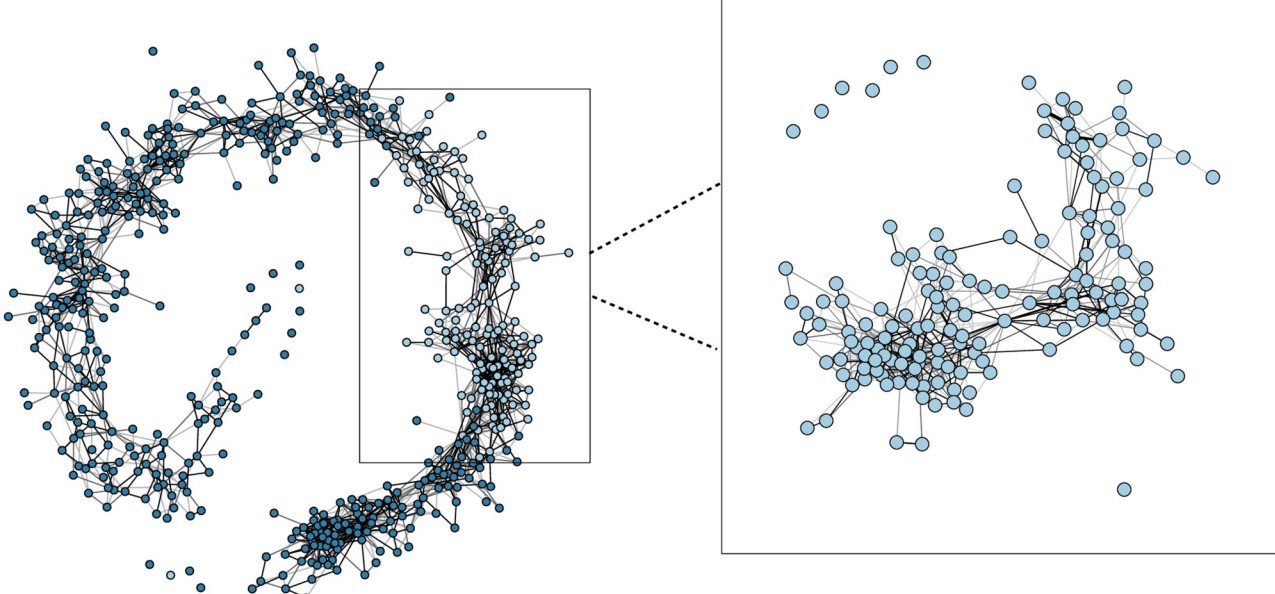

**Fig 2. Contact network in the supermarket.** The contact network of $n$ = 624 participants on March 17[th] is shown on the left. All participants are represented as nodes, and two participants are linked when they came within 1.5 m. The links are weighted by their contact duration. The highlighted nodes indicate the participants present between 15:00 and 16:00, the time slot we selected for the comparison. A detailed view of the contact network of these included participants is shown on the right.

### Interventions

**Reward.** To examine the psychological effect of rewarding participants on their physical distancing behaviour, we compared the contact network on day 6 (no reward) with the contact network of day 7 (reward). As can be seen in Table 2 and Fig 3, participants who received a reward had a median of 6 unique contacts, whereas without a reward participants had a median of 8 unique contacts. Detailed analysis taking the network structure of the data into account showed that the probability of forming contacts was lower when participants received a reward (OR = 0.83, 95% Credible Interval (CI) [0.71, 0.97]). The CI indicates some uncertainty about the size of the effect, but rewarding participants for their effort to keep a distance improved physical distancing. In addition, participants who received a reward had slightly shorter contacts (median of 4.2 seconds) than participants who did not receive a reward (median of 5.9 seconds; $U$ = 21606, $p < 0.001$). Finally, participants' ratings of the regulations, pleasantness, and help did not differ between the two conditions (all $p > 0.2$).

**Signage.** To examine the effect of signage on physical distancing, we compared the contact network on day 3 without signage, to the contact network on day 6, when footprints and arrows were provided. The median number of unique contacts of participants was 9 without signage, and 6 when signage was provided. The probability of participants forming contacts was lower when providing signage compared to the situation where no signage was provided (OR = 0.85, 95% CI [0.71, 1.00]). The CI around this estimate shows that there is some uncertainty about the size of the effect and the upper limit is equal to 1, but indicates that signage is likely to have a positive effect on physical distancing. Finally, contact duration did not differ significantly between conditions ($U$ = 13038, $p$ = 0.07), but participants in the signage condition rated their experience regarding regulations ($t$(383.24) = 2.18, $p$ = .03), pleasantness ($t$(375.51) = 3.41, $p$ <.001) and help ($t$(373.54) = 2.54, $p$ = .01) more satisfactory than participants in the no signage condition.

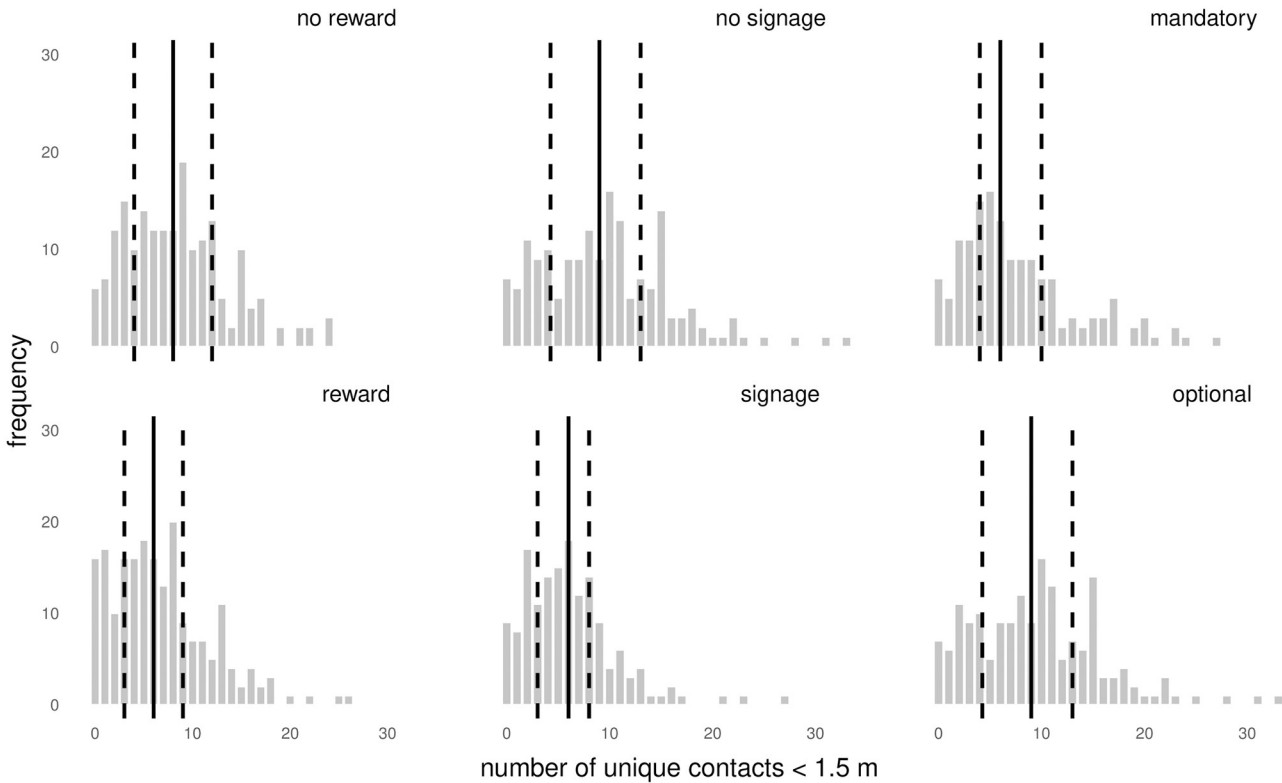

**Fig 3. Contacts per experimental condition.** The number of unique contacts (<1.5 m) in each of the six conditions. The solid line represents the median, and the two dashed lines the 25th and 75th percentiles, such that 50% of the observations fall within the two dashed lines.

**Shopping cart.** To examine the effect of altering shopping cart regulations, we compared the contact network on day 1, when a shopping cart was mandatory, with the contact networks on day 3, when the shopping cart was optional. On day 1 with mandatory shopping carts, participants had a median of 6 unique contacts, compared with a median of 9 unique contacts on day 3, when the shopping carts were optional. Despite these numerical differences, detailed analyses taking the network structure into account indicate that the probability of participants forming contacts was about the same in these two conditions (OR = 1.07, 95% CI [0.90, 1.26]), indicating that if the number of contacts are different between mandatory and optional shopping carts, these differences are likely to be small. Thus, mandatory shopping carts do not appear to facilitate nor inhibit physical distancing. In addition, the contact duration did not differ significantly between conditions ($U = 11724$, $p = 0.68$), and mandatory or optional shopping carts did not change participants' ratings (all $p > 0.1$).

## Discussion

In this paper we aspired to use behavioural science in response to crises by evaluating the effectiveness of interventions on directly observed behaviour. We applied this idea to the COVID-19 crisis and performed a behavioural experiment to investigate the effectiveness of behavioural interventions to promote physical distancing. We did so by implementing three interventions (i.e., reward, signage, shopping cart regulations) and evaluated their effect on the contacts between customers (i.e., a distance within 1.5 m). Our results demonstrate that rewarding customers for keeping their distance and providing signage both improved physical distancing and reduced the number of contacts. Interestingly, rewarding customers not only

reduced the number of contacts, but also shortened the duration of the remaining contacts. In contrast, mandatory or optional shopping cards did not appear to improve or worsen physical distance behaviour. Overall, participants rated their shopping experience as satisfactory, and most of the regulations did not impact these ratings. Only signage was rated more positively on all accounts (regulations, pleasantness, perceived help) than no signage.

The current study showed how we can use behavioural science to find practical behavioural solutions to crises by formulating six key steps. First, starting out with an existing problem, we defined a *desired behaviour* to combat this problem. Second, and importantly, we found a way to *directly* measure this behaviour such that we can evaluate the effectiveness of interventions on a directly observed behavioural outcome measure. Third, based on the desired behaviour, we identified different *interventions* designed to stimulate this behaviour. These interventions can be based on psychological mechanisms (e.g., rewarding participants), but can also be informed by other fields, like in our case crowd management. Ultimately, the aim is to promote the desired behaviour, and any intervention targeted at this can be tested within the *experimental design*. Fourth, in an experimental design we systematically varied the interventions, such that we could in a fifth step *analyse* the effect of each of the interventions on the desired behaviour. Investigating the identified interventions in an experimental design is a crucial step, as behavioural solutions have to be tested (in the crisis situation) before they can be translated into policy [39]. Sixth and finally, the insights derived from the experiment can directly inform policy recommendations, making behavioural science a central link connecting science with society.

In Table 3 we outline the steps described above with concrete examples that we took in identifying behavioural solutions to COVID-19. Crucially, these steps transcend the COVID-19 crisis and can be applied much broader to other crises as well. For the financial crisis as well as the climate crisis these steps can similarly shed light on possible practical behavioural solutions for which we give an illustrative example and accompanying reference in Table 3. These are just a few examples, and the (behavioural) factors involved in these crises are much more complicated than can be captured in a simple table. In addition, in our current experiment we primarily focused on the context in which behaviour occurs, but clearly there are much more factors that influence behaviour such as biology and cognitive processes, social influences, and culture. Nonetheless, the steps that we outlined, and particularly directly observing behaviour, can serve as an avenue to use behavioural change as solution out of a crisis.

While our experimental study provides indications for practical behavioural solutions, there are some limitations that warrant attention. First, because of the naturalistic nature of the study it was not possible to randomize the participants over experimental conditions. As a result, there are several factors that we could not control, like the type of customers or the

**Table 3. Examples on how behavioural science can be used to develop effective interventions to stimulate desired behaviour.**

|  | COVID-19 crisis | Global Financial Crisis | Climate crisis |
|---|---|---|---|
| **problem desired behaviour** | virus transmission physical distancing | housing bubble reduce risk taking | greenhouse gas emission increase vegetarian diets |
| **direct measurement** | physical distances between people measured using UWB | propensity to sell assets | number of vegetarian dishes sold |
| **intervention** | psychological mechanism: reward [28] pedestrian behaviour: follow signage [29] practical: adjust shopping cart regulations | psychological mechanism: salience [40], disposition effect [16] | psychological mechanism: salience [40] practical: visibility |
| **experimental design and analysis** | see current paper | see Frydman & Rangel (2014) [41] | see Kurz (2018) [42] |
| **policy recommendation** | reward people for keeping their distance | decrease salience of information related to capital gains | increase salience of vegetarian dishes |

number of customers in each experimental condition. To limit this variation, we measured during the same days over a two-week time period. In addition, to compare the effectiveness of interventions we selected times during the day that were similar in crowdedness and time of day. Still, it could be possible that some interventions work differently depending on the crowdedness. For example, mandatory shopping carts could potentially facilitate physical distancing in quiet times, but actually crowd the supermarket even more during rush-hours. In the current naturalistic experiment, these factors are hard to disentangle.

Second, it might be challenging to implement some of our findings into practice. For example, we showed that rewarding participants improved physical distancing, but it might be challenging to implement these rewards structurally as supermarkets are frequently visited places. The one-off rewards could be promising for locations that people visit less often (e.g., a cinema or festival), but a different reward scheme might be necessary to achieve long term effects in a supermarket. Third, we did not include any assessments on whether the interventions were adhered to (e.g., whether participants followed the signage or refrained from using a shopping cart). However, we choose to limit the number of questions to maximize the number of participants completing the questionnaire. Even so, only between 25–51% of participants completed the questionnaires, possibly introducing some selection bias. Fourth, since the actual virus spread depends on more than behaviour alone, our study could be extended by collaborating with epidemiologists to quantify the reduction in risk of spread in each of the scenarios.

Last, all behavioural interventions we investigated were implemented in the context of a supermarket. To translate these findings to other situations it is important to consider relevant characteristics of different contexts. For example, in supermarkets people are likely to move constantly, and often visit a fixed set of locations corresponding to one's shopping list. These characteristics may be substantially different from other situations, such as using public transport (where people may have to wait for their train to arrive), or visiting a museum (where people often visit an entire exhibition). In a previous study we showed that walking directions also facilitated physical distancing at an art fair [27], indicating that walking directions may be applicable both in situations where people plan their own stops (as in the supermarket) and in situations where people follow pre-specified routes and stops (as in the art fair).

## Conclusion

To conclude, in this paper we have shown how we can find practical behavioural solutions to a crisis by defining and directly observing the desired behaviour in an experimental design. Of course, behavioural science alone will not offer the complete package to combat an entire crisis by itself. We need multidisciplinary collaborations to battle the multifaceted and complex problems of our time [22]. In these collaborations, behavioural science and behavioural change can provide new ways to look at existing (and new) challenges. We should start to take advantage of the opportunities offered by behavioural science.

## Supporting information

**S1 Appendix. Information flyer.** The translated flyer (originally Dutch) that was distributed to inform people of the upcoming experiment in the supermarket.
(PDF)

## Acknowledgments

We would like to thank André and Joyce van Reijen for opening up their supermarket to run this experiment, and PLUS for printing all signage. We thank our team that helped collecting

the data: Frederike Meijer, Sonja van Meerbeek, Zuzana Wilms, Henk Nieweg, Nina Leach, and Lander Arteaga.

## Author Contributions

**Conceptualization:** Charlotte C. Tanis, Meier J. Boersma, Maya V. Van der Steenhoven, Denny Borsboom, Tessa F. Blanken.

**Data curation:** Charlotte C. Tanis, Floor H. Nauta, Tessa F. Blanken.

**Formal analysis:** Charlotte C. Tanis, Floor H. Nauta, Denny Borsboom, Tessa F. Blanken.

**Funding acquisition:** Meier J. Boersma, Maya V. Van der Steenhoven, Denny Borsboom.

**Investigation:** Charlotte C. Tanis, Floor H. Nauta, Meier J. Boersma, Maya V. Van der Steenhoven, Denny Borsboom, Tessa F. Blanken.

**Methodology:** Charlotte C. Tanis, Floor H. Nauta, Denny Borsboom, Tessa F. Blanken.

**Project administration:** Charlotte C. Tanis, Floor H. Nauta, Meier J. Boersma, Tessa F. Blanken.

**Resources:** Meier J. Boersma, Maya V. Van der Steenhoven.

**Supervision:** Meier J. Boersma, Tessa F. Blanken.

**Validation:** Charlotte C. Tanis, Floor H. Nauta, Tessa F. Blanken.

**Visualization:** Charlotte C. Tanis, Floor H. Nauta, Tessa F. Blanken.

**Writing – original draft:** Charlotte C. Tanis, Tessa F. Blanken.

**Writing – review & editing:** Charlotte C. Tanis, Floor H. Nauta, Meier J. Boersma, Maya V. Van der Steenhoven, Denny Borsboom, Tessa F. Blanken.

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
