## [Decision Letter · Decision Letter 0]

28 Mar 2022

PONE-D-21-32828Practical behavioural solutions to COVID-19: Changing the role of behavioural science in crisesPLOS ONE

Dear Dr. Tanis,

Thank you for submitting your manuscript to PLOS ONE. After careful consideration, we feel that it has merit but does not fully meet PLOS ONE’s publication criteria as it currently stands. Therefore, we invite you to submit a revised version of the manuscript that addresses the points raised during the review process.

ACADEMIC EDITOR: I have now received sufficient review reports. Both reviewers recommended publication but also suggested revisions to your manuscript, especially on the methodology section where authors need to clarify certain issues highlighted by the reviewers.

We look forward to receiving your revised manuscript.

Kind regards,

Gabriel Hoh Teck Ling, PhD

Academic Editor

PLOS ONE

“We would like to thank André and Joyce van Reijen for opening up their supermarket to run this experiment, and PLUS for

printing all signage. We thank our team that helped collecting the data: Frederike Meijer, Sonja van Meerbeek, Zuzana Wilms,

Henk Nieweg, Nina Leach, and Lander Arteaga. The research project was supported by the Ministry of Economic Affairs and

Climate Policy. CT and TB were supported by an Innovation Exchange Amsterdam UvA Proof of Concept Fund”

“The research project was supported by the Ministry of Economic Affairs and Climate Policy. CT and TB were supported by an Innovation Exchange Amsterdam UvA Proof of Concept Fund. The funders had no role in study design, data collection and analysis, decision to publish, or preparation of the manuscript.”

Reviewers' comments:

Reviewer's Responses to Questions

**Comments to the Author**

1. Is the manuscript technically sound, and do the data support the conclusions?

Reviewer #1: Yes

Reviewer #2: Partly

2. Has the statistical analysis been performed appropriately and rigorously? 

Reviewer #1: I Don't Know

Reviewer #2: Yes

3. Have the authors made all data underlying the findings in their manuscript fully available?

Reviewer #1: Yes

Reviewer #2: No

4. Is the manuscript presented in an intelligible fashion and written in standard English?

Reviewer #1: Yes

Reviewer #2: Yes

5. Review Comments to the Author

Reviewer #1: This is an interesting experimental study examining the impact of rewards and cues, one component of behavioural interventions, on distancing behaviours inn a supermarket in the Netherlands during the initial wave of the COVID-19 pandemic. The study was well done and the analyses appear sound. I do not feel sufficiently expert to comment on the statistical approaches used; a statistical reviewer might be helpful.

While I do like this paper, I think that it overstates its purpose. A reader not familiar with behavioural science would conclude that this study is groundbreaking in introducing behavioural sciences to crises. This is simply not true and misleading on 2 counts. First, behavioural sciences have been quite active in the pandemic response. I did a quick PubMed search with the search terms "behavioural science" and "COVID" and got 106 papers from 2020 - 2022, including the following papers directly on behavioural science approaches:

Using social and behavioural science to support COVID-19 pandemic response

Mental health and clinical psychological science in the time of COVID-19: Challenges, opportunities, and a call to action

Research priorities for the COVID-19 pandemic and beyond: A call to action for psychological science

Psychological science and COVID-19: An agenda for social action

Infected by Bias: Behavioral Science and the Legal Response to COVID-19

Trust in Science, Perceived Media Exaggeration About COVID-19, and Social Distancing Behavior

The cognitive science of COVID-19: Acceptance, denial, and belief change

Intervening on Trust in Science to Reduce Belief in COVID-19 Misinformation and Increase COVID-19

Preventive Behavioral Intentions: Randomized Controlled Trial

Ending the Pandemic: How Behavioural Science Can Help Optimize Global COVID-19 Vaccine Uptake

Applying relationship science to evaluate how the COVID-19 pandemic may impact couples' relationships

Individual health behaviours to combat the COVID-19 pandemic: lessons from HIV socio-behavioural science

How behavioural science data helps mitigate the COVID-19 crisis

Lessons From the UK's Lockdown: Discourse on Behavioural Science in Times of COVID-19

Covid-19: What we have learnt from behavioural science during the pandemic so far that can help prepare us for the future

The Science of Persuasion Offers Lessons for COVID-19 Prevention

Can Behavioral Science Help Us Fight COVID-19

Fear of Covid-19: Insights from Evolutionary Behavioral Science

Process-based functional analysis can help behavioral science step up to novel challenges: COVID - 19 as an example

The Dynamics of Fear at the Time of Covid-19: A Contextual Behavioral Science Perspective

Effect of Targeted Behavioral Science Messages on COVID-19 Vaccination Registration Among Employees of a Large Health System: A Randomized Trial

Harnessing behavioural science in public health campaigns to maintain 'social distancing' in response to the COVID-19 pandemic: key principles

It falls to the authors to place their study into the context of the behavioural sciences. Second, the authors focus on one very specific aspect of the behavioural sciences; environmental context. They, in essence, draw on the sub-area of behaviour modification, primarily using cues and rewards to shape behaviour. There is nothing wrong with this, except behavioural science is broader, to include social influences, culture, biology, and the full range of cognitive processing characteristics. Again, nothing wrong with what the authors have done but they should inform the readers of the specific aspects of the behavioural science approach they are taking. For instance, great gains have been made by framing behavioural sciences within what is called the Theoretical Domains Framework, an integration of 33 behavioural change theories, that has identified 14 domains of behaviour change intervention and has been effectively summarized with the COM-B model; behaviour is the result of Capability, Opportunity, and Motivation. This paper falls within the Opportunity domain. For the author' information, this model has been developed from University College London UK, under the guidance of the behavioural scientist Dr. Susan Michie, who is a member of the UK COVID Response team at the highest level of government (the point being it is not accurate to say behavioural science has been left out of the response to COVID - other countries also have behavioural science teams offering advice).

The methodology of this study is very interesting and appears sound. I can see how this methodology can be useful for specific questions, I am a bit confused, however, by Table 1. We see 7 days of intervention but the analyses only involve comparing days 1, 3 (Shopping cart), 6 (signage) and 7 (rewards). What is the purpose of days 2, 4, 5? On that note the Table lists 'space' as an intervention but this is labelled shopping cart in the text.

In the discussion I wonder if the authors have any comment about this study being conducted at the beginning of lockdown experience, where most of the population was experiencing threat. Now that we are almost 2 years in, and many in the population are experiencing demoralization of outrage (the Netherlands has made international coverage of protests recently) do the authors think the study would yield the same results?

I look forward to the contribution of this work to the field, once the study is appropriately contextualized.

Reviewer #2: Summary

This was an interesting and timely naturalistic experimental study that examined the efficacy of three behavioural interventions for promoting physical distancing behaviour in grocery stores during the covid-19 pandemic: (i) rewarding customers for keeping distance; (i) providing signage to guide customers; and (iii) altering shopping cart regulations. They recruited 4323 participants and the main outcome was number of contacts less than 1.5 between customers measured using network analysis. Results showed that both rewards and signage increased physical distancing, but shopping cart regulations did not. Rewards also reduced the duration of contacts. The authors concluded that incorporating behavioural science approaches and interventions into pandemic management should be strengthened and emphasized to improve pandemic outcomes.

Comments

• The introduction of this paper was very compelling – the fact that in times of crisis we turn to crisis-specific experts (e.g., economists during financial crises), the authors did an outstanding job of asking why, given the importance of engaging in preventive behaviours (from distancing to vaccination) during the covid-19 pandemic crisis, did we not turn to behavioural science experts?

• The study was also generally well reasoned in terms of exploring the efficacy of different behavioural interventions to promote distancing behaviour. However, the choice of specific interventions was not described or articulated. Authors did not justify their underlying theoretical rationale (why would they be expected to change behaviour in this context and why these interventions over others?). Tying each intervention to an established behaviour change theory or model would strengthen the paper and highlight the importance of doing this in general. For example, the rationale for the shopping cart intervention is not obvious to me.

• Could the authors clarify what participants were told about the objective of the study – for example: did they know what each intervention was and what outcome was being measured? The authors described this as a naturalistic experiment, but if they knew what was being measured and why, this could have influenced their behaviour more than just being exposed to the intervention (without details).

• Could the authors also clarify if they delivered the interventions the same way they would have been delivered were they implemented in ‘real life’? For example, there were study personnel present to explain the study, hand out tags, and answer questions. Would these resources be available if we were deliver the interventions in real life? Would these roles be assumed by store personnel? The use of an implementation science approach to intervention design an delivery was not explicit.

• The use of objective measures of distancing (SafeTag) was judged to be a strength based on the non-intrusiveness and validity of the measures.

• The authors described how they treated shoppers who were shopping together (as they would likely have close contact throughout the intervention that needed to be accounted for). They described how they accounted for this (based on contact metrics), but the potential for misclassification seems high. For example, many people or families shopping together ‘split up’ in the interests of time – these shopping patterns may have been miscalculated for these groups. Why not just ‘tag’ people shopping together when they enter the store and receive their tags, so irrespective of their shopping patterns, they would not be counted in distancing measures (because we don’t expect those living together or family groups to distance). Could the authors clarify this?

• Given that national regulations at the time of the study limited the number of customers in the store (max one customer per 10m2), store access had to be monitored and controlled. The authors seemed to account for this by comparing conditions at times that were similar in terms of crowdedness, which is appropriate.

• For the signage intervention, the authors did not appear to assess how many people viewed the signs (eg, during exit interviews or surveys). It is difficult to attribute behavioural changes to this intervention in the absence of verifying the extent to which the intervention was ‘received’ by shoppers.

• The authors conducted experience assessments (though only 25-51% completed them), though these assessments did not appear to validate receipt of the interventions (e.g., viewing the signs) or the extent to which the decision to maintain distancing with automatic or reflexive (as per COM-B model)? This would have pointed to the mechanism of action of the interventions (which is how we think interventions are working), and so not assessing these things seems like a missed opportunity.

• The discussion could be strengthened by a discussion of the effect size of their findings and the extent to which results, some of which were statistically significant, were also clinically significant in terms of ability to translate into reduction of virus transmission. For example, on a population level, is reducing contacts from 8 to 6 clinically significant? Is reducing contact time by 1.7 seconds enough to prevent virus transmission? While I wholeheartedly agree that more behavioural interventions need to be designed, evaluated and implemented, we need to ensure we are demonstrating the value of these interventions for addressing the crisis at hand, which is going to strengthen the credibility of the argument to turn to behavioural science for solutions to the pandemic crisis.

6. PLOS authors have the option to publish the peer review history of their article (what does this mean?). If published, this will include your full peer review and any attached files.

Reviewer #1: No

Reviewer #2: **Yes: **Kim L. Lavoie

---

## [Author Response · Author response to Decision Letter 0]

15 Jun 2022

Note: mark-up was lost when copying the text in this field. A version in which we added mark-up to facilitate the review process has been added as file.

Reviewer #1: This is an interesting experimental study examining the impact of rewards and cues, one component of behavioural interventions, on distancing behaviours inn a supermarket in the Netherlands during the initial wave of the COVID-19 pandemic. The study was well done and the analyses appear sound. I do not feel sufficiently expert to comment on the statistical approaches used; a statistical reviewer might be helpful.

While I do like this paper, I think that it overstates its purpose. A reader not familiar with behavioural science would conclude that this study is groundbreaking in introducing behavioural sciences to crises. This is simply not true and misleading on 2 counts. First, behavioural sciences have been quite active in the pandemic response. I did a quick PubMed search with the search terms "behavioural science" and "COVID" and got 106 papers from 2020 - 2022, including the following papers directly on behavioural science approaches: <references>. It falls to the authors to place their study into the context of the behavioural sciences. 

We thank the reviewer for raising this important point and list of references. Upon rereading the manuscript with this comment in mind, we understand that it might be read as if we aim to introduce behavioural science to crisis, so we very much agree that we should stress the contributions of behavioural science to crises more, as we absolutely do not wish to claim that we are the first ones to do so. We do argue that we can utilize behavioural science more, specifically by investigating the effectiveness of interventions on *directly observed behaviour*. To place our study in the context of the behavioural sciences, we have now added additional references, especially throughout the introduction, and make a clearer distinction on what has already been done and what is new in this manuscript. See for example page 2 lines 16-19:

“This need for behavioural change resulted in numerous studies into factors that determine behaviour [8–10], and worldwide regulations such as lockdowns, school closures, and travel restrictions, all directed to reduce the number of behavioural contacts.”

Second, the authors focus on one very specific aspect of the behavioural sciences; environmental context. They, in essence, draw on the sub-area of behaviour modification, primarily using cues and rewards to shape behaviour. There is nothing wrong with this, except behavioural science is broader, to include social influences, culture, biology, and the full range of cognitive processing characteristics. Again, nothing wrong with what the authors have done but they should inform the readers of the specific aspects of the behavioural science approach they are taking. For instance, great gains have been made by framing behavioural sciences within what is called the Theoretical Domains Framework, an integration of 33 behavioural change theories, that has identified 14 domains of behaviour change intervention and has been effectively summarized with the COM-B model; behaviour is the result of Capability, Opportunity, and Motivation. This paper falls within the Opportunity domain. For the author' information, this model has been developed from University College London UK, under the guidance of the behavioural scientist Dr. Susan Michie, who is a member of the UK COVID Response team at the highest level of government (the point being it is not accurate to say behavioural science has been left out of the response to COVID - other countries also have behavioural science teams offering advice).

We thank the reviewer for this excellent comment. Again, we never meant to imply that behavioural science was left out of the response to COVID-19 altogether. Additionally, we agree that the manuscript is improved by placing this study within the COM-B model. We have now included an entire paragraph on the COM-B model in the introduction:

Behavioural science offers many models into the determinants of behaviour, and thereby offers leveraging points on how to instantiate behavioural change. The widely used Capability, Opportunity, Motivation, Behaviour (COM-B) model, for example, posits that a particular behaviour occurs when someone has the capability (i.e., psychological and physical), opportunity (i.e., contextual factors that facilitate the behaviour), and motivation (i.e., processes that energize and direct behaviour) to enact the behaviour [21]. By outlining the factors that influence particular behaviours to occur, the model also provides the opportunity to identify different points of engagement to bring about the desired behavioural change.

We also added the following sentence in the discussion:

In addition, in our current experiment we primarily focused on the context in which behaviour occurs, but clearly there are much more factors that influence behaviour such as biology and cognitive processes, social influences, and culture.

The methodology of this study is very interesting and appears sound. I can see how this methodology can be useful for specific questions, I am a bit confused, however, by Table 1. We see 7 days of intervention but the analyses only involve comparing days 1, 3 (Shopping cart), 6 (signage) and 7 (rewards). What is the purpose of days 2, 4, 5? On that note the Table lists 'space' as an intervention but this is labelled shopping cart in the text.

To isolate the effect of each intervention, we chose to compare days that only differed on that factor while keeping the remaining factors constant. In the design of the study, we build in some redundancy so that we could minimise the effect of factors such as day of the week and time of day, which are likely to have an effect on the type of customers present in the supermarket and could, as such, potentially affect the comparisons that were made. . Due to this redundancy, we did not actually include all days in our analysis. To clarify these decisions, we have now added the following text when describing the table:

Note that there is some redundancy in the experimental conditions we implemented: for example, the effect of reward could be assessed by comparing day four to day three or two, but also by comparing day 7 to day 6. We did so to minimise the influence of factors such as day and time of day. Both the day and time of day are likely to affect the type of customers and crowdedness in the supermarket. Customers during workdays may differ from customers during the weekend, and customers during the morning may differ from customers after work-hours. Similarly, the crowdedness in the supermarket is likely to affect the number of contacts made. Since these factors are difficult to control in a naturalistic study, we implemented some redundancy in the design so that we could select time intervals post-hoc that would minimise differences in these factors for an optimal comparison across conditions. 

In addition, we extended the following sentences in the Descriptive analyses section:

To evaluate the effectiveness of each intervention in isolation, we select two conditions that differ only in regard to whether the behavioural intervention of interest is implemented and that are similar in crowdedness and time of day. This way we can isolate the effect of the intervention while keeping other factors constant.

Finally, we thank the reviewer for pointing out the inconsistency in the phrasing of “shopping cart” and “space”. We have updated the text in the table to also refer to this intervention as “shopping cart”.

In the discussion I wonder if the authors have any comment about this study being conducted at the beginning of lockdown experience, where most of the population was experiencing threat. Now that we are almost 2 years in, and many in the population are experiencing demoralization of outrage (the Netherlands has made international coverage of protests recently) do the authors think the study would yield the same results?

This is an interesting point and upon careful consideration we actually think that the study would yield similar results. Specifically, when we conducted the study in March 2021, we were already one year into the pandemic. During that time there were some large-scale demonstrations (e.g., there were some severe curfew riots in The Netherlands January 23-26 2021) and demorilazation due to ongoing lockdowns. So, we do think that results between March 2021 and later on in the pandemic would be more comparable than if our study was conducted right at the start of the pandemic. Moreover, the effectiveness of the reward intervention also fits in a larger psychological context, where threat does not need to be present for rewards to promote desired behaviour.

I look forward to the contribution of this work to the field, once the study is appropriately contextualized.

Reviewer #2: Summary

This was an interesting and timely naturalistic experimental study that examined the efficacy of three behavioural interventions for promoting physical distancing behaviour in grocery stores during the covid-19 pandemic: (i) rewarding customers for keeping distance; (i) providing signage to guide customers; and (iii) altering shopping cart regulations. They recruited 4323 participants and the main outcome was number of contacts less than 1.5 between customers measured using network analysis. Results showed that both rewards and signage increased physical distancing, but shopping cart regulations did not. Rewards also reduced the duration of contacts. The authors concluded that incorporating behavioural science approaches and interventions into pandemic management should be strengthened and emphasized to improve pandemic outcomes.

Comments

• The introduction of this paper was very compelling – the fact that in times of crisis we turn to crisis-specific experts (e.g., economists during financial crises), the authors did an outstanding job of asking why, given the importance of engaging in preventive behaviours (from distancing to vaccination) during the covid-19 pandemic crisis, did we not turn to behavioural science experts?

We thank the reviewer for judging the introduction as compelling.

• The study was also generally well reasoned in terms of exploring the efficacy of different behavioural interventions to promote distancing behaviour. However, the choice of specific interventions was not described or articulated. Authors did not justify their underlying theoretical rationale (why would they be expected to change behaviour in this context and why these interventions over others?). Tying each intervention to an established behaviour change theory or model would strengthen the paper and highlight the importance of doing this in general. For example, the rationale for the shopping cart intervention is not obvious to me.

We agree with the reviewer that the paper is strengthened by providing the rationale of choosing our set of interventions. We have now extended the following paragraph in the Participants and Design section to elaborate on our choices:

The experiment took place during seven days in a supermarket. We varied three interventions: reward [28], signage [29], and adjusting the shopping cart regulations. These interventions were chosen for varying reasons. First, rewarding people for displaying the desired behaviour is well-established to be effective in promoting that desired behaviour [30], and also advised during the COVID-19 pandemic in particular [31]. In our study, participants received the reward upon handing in their tag (see Procedure) and consisted of cookies on Saturday March 24th and chocolate on Friday March 26th. Second, signage is commonly used to change behaviour [29] and often used in traffic to, for example, avoid collisions [32]. We aimed to investigate whether clear signage would facilitate pedestrian flows and thereby physical distancing. We included arrows signaling unidirectional walking directions in part of the supermarket and footprints in the queue for the register (see Materials below). Third, we changed the shopping cart regulations from mandatory (which was standard in the Netherlands at that time) to optional. The mandatory shopping carts were implemented in the Netherlands in March 2020 [33] to (1) keep track of the number of participants in the supermarket as national regulations allowed a maximum of 1 customer per 10 m2, and (2) as the shopping carts were thought to facilitate physical distancing. At the same time it could be argued that the mandatory shopping carts take up a lot of space within the supermarket and, as such, hinder the opportunity for physical distancing. 

• Could the authors clarify what participants were told about the objective of the study – for example: did they know what each intervention was and what outcome was being measured? The authors described this as a naturalistic experiment, but if they knew what was being measured and why, this could have influenced their behaviour more than just being exposed to the intervention (without details).

We thank the reviewer for raising this point. To elaborate on which information was provided to participants we have now added the information flyer as Supporting Information. The participants were aware that we measured distance to other visitors in the supermarket. Thus, it is likely that handing out the sensors (without any additional intervention) already has an effect on the distance that visitors maintained while shopping. Therefore we included two baseline conditions (Wednesday 17 March, and 24 March) in which we merely handed out the sensors, without any additional intervention. This allows us to evaluate the effect of a single intervention, while taking the potential effect of measuring the behaviour (also known as ‘mere measurement effect’) into account. 

• Could the authors also clarify if they delivered the interventions the same way they would have been delivered were they implemented in ‘real life’? For example, there were study personnel present to explain the study, hand out tags, and answer questions. Would these resources be available if we were deliver the interventions in real life? Would these roles be assumed by store personnel? The use of an implementation science approach to intervention design an delivery was not explicit.

Extra personnel was only present to hand out and hand in the sensors, but the shopping experience itself was the same as usual. If evidence was found in favour of the effectiveness of an intervention, that intervention could then be implemented after the conclusion of the study. At that point, no sensors (or extra personnel) would be needed. For example, finding that providing signage works, then signage could be placed inside the store without needing to measure distances again. To clarify how the interventions were implemented, and which extra personnel was present we added the following to the Procedures section:

In conditions in which a reward was handed out, we also informed participants that they would receive a reward for their effort to keep their distance upon handing in their SafeTag. Participants could then proceed to do their groceries like they would otherwise do. Thus, as soon as the participants entered the supermarket they did not encounter any study personnel.

• The use of objective measures of distancing (SafeTag) was judged to be a strength based on the non-intrusiveness and validity of the measures.

We thank the reviewer for this comment and we agree that the SafeTag is a great non-intrusive way to measure distance. Another option would be to use cameras, but that brings quite some privacy issues. Using the SafeTags, we never need to know which person wears which tag.

• The authors described how they treated shoppers who were shopping together (as they would likely have close contact throughout the intervention that needed to be accounted for). They described how they accounted for this (based on contact metrics), but the potential for misclassification seems high. For example, many people or families shopping together ‘split up’ in the interests of time – these shopping patterns may have been miscalculated for these groups. Why not just ‘tag’ people shopping together when they enter the store and receive their tags, so irrespective of their shopping patterns, they would not be counted in distancing measures (because we don’t expect those living together or family groups to distance). Could the authors clarify this?

We understand that the rules we have for tagging people as belonging from the same household might seem quite cumbersome. The reason why we chose this approach was that we needed to be able to hand out the sensors as quickly as possible. Especially during COVID-19, we could not risk creating congestion at the entrance of the store. For that reason, we did not have time to register upon entering which visitors came in together. Whenever two participants were tagged as belonging to the same group, only the contacts between them were not counted in analysis. Any contact made outside of the group was included. To clarify these choices we have now included the following in the Procedure section:

If customers of the same household participated, they each received a SafeTag. According to regulations, household members did not need to keep a distance, but we were unable to register their group membership upon handing out the tag to avoid congestion. We did exclude contacts between group members when processing the data and only kept contacts with individuals outside of the group (see Pre-processing).

In addition we expanded the Pre-processing section:

We then checked if participants entered the supermarket as a household by investigating three criteria: at least 10 contacts within 80 cm, being within 1.5 m of each other for at least 25% of their visit duration, and exiting the supermarket at most 60 seconds after each other. We assigned participants as belonging to the same group if they met at least two of these criteria and removed all contacts between the respective group members.

• Given that national regulations at the time of the study limited the number of customers in the store (max one customer per 10m2), store access had to be monitored and controlled. The authors seemed to account for this by comparing conditions at times that were similar in terms of crowdedness, which is appropriate.

We thank the reviewer for appreciating the way in which we chose to compare conditions.

• For the signage intervention, the authors did not appear to assess how many people viewed the signs (eg, during exit interviews or surveys). It is difficult to attribute behavioural changes to this intervention in the absence of verifying the extent to which the intervention was ‘received’ by shoppers.

• The authors conducted experience assessments (though only 25-51% completed them), though these assessments did not appear to validate receipt of the interventions (e.g., viewing the signs) or the extent to which the decision to maintain distancing with automatic or reflexive (as per COM-B model)? This would have pointed to the mechanism of action of the interventions (which is how we think interventions are working), and so not assessing these things seems like a missed opportunity.

We agree that it would have been ideal to also assess how many people viewed the signs. We have now included this point as a limitation in the Discussion section:

Third, we did not include any assessments on whether the interventions were adhered to (e.g., whether participants followed the signage or refrained from using a shopping cart). However, we choose to limit the number of questions to maximize the number of participants completing the questionnaire. Even so, only between 25-51% of participants completed the questionnaires, possibly introducing some selection bias.

• The discussion could be strengthened by a discussion of the effect size of their findings and the extent to which results, some of which were statistically significant, were also clinically significant in terms of ability to translate into reduction of virus transmission. For example, on a population level, is reducing contacts from 8 to 6 clinically significant? Is reducing contact time by 1.7 seconds enough to prevent virus transmission? While I wholeheartedly agree that more behavioural interventions need to be designed, evaluated and implemented, we need to ensure we are demonstrating the value of these interventions for addressing the crisis at hand, which is going to strengthen the credibility of the argument to turn to behavioural science for solutions to the pandemic crisis.

We again agree with the reviewer that this is a very interesting point. As we do not ourselves have a background in epidemiology we feel uncomfortable to directly translate these effects into clinically meaningful statements. However, we have now included the following as a limitation in the Discussion section:

Fourth, since the actual virus spread depends on more than behaviour alone, our study could be extended by collaborating with epidemiologists to quantify the reduction in risk of spread in each of the scenarios.

---

## [Decision Letter · Decision Letter 1]

1 Aug 2022

Practical behavioural solutions to COVID-19: Changing the role of behavioural science in crises

PONE-D-21-32828R1

Dear Dr. Tanis,

We’re pleased to inform you that your manuscript has been judged scientifically suitable for publication and will be formally accepted for publication once it meets all outstanding technical requirements.

Kind regards,

Gabriel Hoh Teck Ling, PhD

Academic Editor

PLOS ONE

Additional Editor Comments (optional):

Reviewers' comments:

Reviewer's Responses to Questions

**Comments to the Author**

1. If the authors have adequately addressed your comments raised in a previous round of review and you feel that this manuscript is now acceptable for publication, you may indicate that here to bypass the “Comments to the Author” section, enter your conflict of interest statement in the “Confidential to Editor” section, and submit your "Accept" recommendation.

Reviewer #1: All comments have been addressed

2. Is the manuscript technically sound, and do the data support the conclusions?

Reviewer #1: Yes

3. Has the statistical analysis been performed appropriately and rigorously? 

Reviewer #1: I Don't Know

4. Have the authors made all data underlying the findings in their manuscript fully available?

Reviewer #1: Yes

5. Is the manuscript presented in an intelligible fashion and written in standard English?

Reviewer #1: Yes

6. Review Comments to the Author

Reviewer #1: (No Response)

7. PLOS authors have the option to publish the peer review history of their article (what does this mean?). If published, this will include your full peer review and any attached files.

Reviewer #1: No

---

## [Editor Report · Acceptance letter]

14 Sep 2022

PONE-D-21-32828R1 

Practical behavioural solutions to COVID-19: Changing the role of behavioural science in crises 

Dear Dr. Tanis:

I'm pleased to inform you that your manuscript has been deemed suitable for publication in PLOS ONE. Congratulations! Your manuscript is now with our production department. 

Kind regards, 

on behalf of

Dr. Gabriel Hoh Teck Ling 

Academic Editor

PLOS ONE